# Wild *Vigna* Legumes: Farmers' Perceptions, Preferences, and Prospective Uses for Human Exploitation

**Difo Voukang Harouna** [1,3], **Pavithravani B. Venkataramana** [2,3], **Athanasia O. Matemu** [1,3] and **Patrick Alois Ndakidemi** [2,3,*]

1   Department of Food Biotechnology and Nutritional Sciences, Nelson Mandela African Institution of Science and Technology (NM-AIST), P.O. Box 447 Arusha, Tanzania; harounad@nm-aist.ac.tz (D.V.H.); athanasia.matemu@nm-aist.ac.tz (A.O.M.)
2   Department of Sustainable Agriculture, Biodiversity and Ecosystems Management, Nelson Mandela African Institution of Science and Technology, P.O. Box 447 Arusha, Tanzania; pavithravani.venkataramana@nm-aist.ac.tz
3   Centre for Research, Agricultural Advancement, Teaching Excellence and Sustainability in Food and Nutrition Security (CREATES-FNS), Nelson Mandela African Institution of Science and Technology, P.O. Box 447 Arusha, Tanzania
*   Correspondence: patrick.ndakidemi@nm-aist.ac.tz; Tel.: +255-757744772

**Abstract:** The insufficient food supply due to low agricultural productivity and quality standards is one of the major modern challenges of global agricultural food production. Advances in conventional breeding and crop domestication have begun to mitigate this issue by increasing varieties and generation of stress-resistant traits. Yet, very few species of legumes have been domesticated and perceived as usable food/feed material, while various wild species remain unknown and underexploited despite the critical global food demand. Besides the existence of a few domesticated species, there is a bottleneck challenge of product acceptability by both farmers and consumers. Therefore, this paper explores farmers' perceptions, preferences, and the possible utilization of some wild *Vigna* species of legumes toward their domestication and exploitation. Quantitative and qualitative surveys were conducted in a mid-altitude agro-ecological zone (Arusha region) and a high altitude agro-ecological zone (Kilimanjaro region) in Tanzania to obtain the opinions of 150 farmers regarding wild legumes and their uses. The study showed that very few farmers in the Arusha (28%) and Kilimanjaro (26%) regions were aware of wild legumes and their uses. The study further revealed through binary logistic regression analysis that the prior knowledge of wild legumes depended mainly on farmers' location and not on their gender, age groups, education level, or farming experience. From the experimental plot with 160 accessions of wild *Vigna* legumes planted and grown up to near complete maturity, 74 accessions of wild *Vigna* legumes attracted the interest of farmers who proposed various uses for each wild accession. A $X^2$ test (likelihood ratio test) revealed that the selection of preferred accessions depended on the farmers' gender, location, and farming experience. Based on their morphological characteristics (leaves, pods, seeds, and general appearance), farmers perceived wild *Vigna* legumes as potentially useful resources that need the attention of researchers. Specifically, wild *Vigna* legumes were perceived as human food, animal feed, medicinal plants, soil enrichment material, and soil erosion-preventing materials. Therefore, it is necessary for the scientific community to consider these lines of farmers' suggestions before carrying out further research on agronomic and nutritional characteristics toward the domestication of these alien species for human exploitation and decision settings.

**Keywords:** non-domesticated legumes; *Vigna racemosa*; *Vigna ambacensis*; *Vigna reticulata*; *Vigna vexillata*; Tanzania; wild food legumes

## 1. Introduction

Legumes (family: Fabaceae) possess an undeniable vital nutritional value for both humans and animals due to their protein content. They are known to be the second most valuable plant source of nutrients for both humans and animals, and the third largest family among flowering plants, with about 650 genera and 20,000 species [1]. Some of the most commonly domesticated, grown, and commercialized legumes such as soybeans, cowpeas, common beans, and other forms have demonstrated considerable contribution to the global food security [2]. Yet, their production rate remains unsatisfying compared with their consumption rate due to biotic and abiotic challenges [3]. Therefore, there is a need to look for alternatives. A systematic screening of the hitherto wild non-domesticated and wild relatives of the domesticated species within the commonly known and the little-known genera of legumes might be a promising strategy.

The *Phaseolus* and *Vigna* genera comprises the most widely consumed legumes, namely common beans (*Phaseolus vulgaris*) and cowpea (*Vigna unguiculata*) [2,4,5]. Within each genus, there are fewer domesticated edible species as compared with the numerous non-domesticated wild species. Some domesticated or semi-domesticated species have been termed as neglected and underutilized species due to little attention being paid to them or the complete ignorance of their existence by agricultural researchers, plant breeders, and policymakers [6]. This study mainly focuses on the genus *Vigna*.

The genus *Vigna* is a huge and important set of legumes consisting of more than 200 species [7]. It comprises several species of agronomic, economic, and environmental importance. The most common domesticated ones include the mung bean [*V. radiata* (L.) Wilczek], urd bean [*V. mungo* (L.) Hepper], cowpea [*V. unguiculata* (L.) Walp.], azuki bean [*V. angularis* (Willd.) Ohwi & Ohashi], bambara groundnut [*V. subterranea* (L.) Verdc.], moth bean [*V. aconitifolia (Jacq.) Maréchal*], and rice bean [*V. umbellata* (Thunb.) Ohwi & Ohashi]. Many of these species are valued as forage, green manure, and cover crops, besides their value as high protein grains. The genus *Vigna* also comprises more than 100 wild species that do not possess common names apart from their scientific appellation yet [8]. They are given different denotations such as underexploited wild *Vigna* species, non-domesticated *Vigna* species, wild *Vigna*, or alien species, depending on the scientist [2,7,9].

The rapid evolution, distribution, and spreading of improved bred crop varieties due to breeding programs and domestication in order to respond to food security challenges have also impacted positively on the negligence and disappearance of wild crop relatives [10,11]. This is certainly a negative impact vis-à-vis the species' biodiversity conservation. From that perspective, one could imagine and question the awareness, beliefs, and preferences of some generations regarding the origin of the consumed modern crops. This may explain the stigma about the consumption and even the existence of these wild legumes, and therefore their rejection as food while they have been used as such in the past in some cases.

Food acceptability and food choices are usually influenced by many factors in which sensory preferences play an important role [12]. The nutritional composition is also a very essential characteristic to consider in food selection and consumption, as it is directly linked to consumers' health and well-being. Unfortunately, this parameter may only be seriously considered in parts of the world where food accessibility, availability, and affordability are not challenged. Hence, much is needed to be done in this line to investigate the nutritional composition of wild crop foods together with close understanding of their social acceptability.

Investigations on the chemical composition of wild *Vigna* legumes seem to be less attractive to the scientific community for reasons yet to be established. Research in that line has remained silent and undocumented for more than a decade [13]. The latest report shows that some of the wild *Vigna*

accessions studied present nutrient levels comparable to those of some domesticated species with exceptionally higher levels of sulfur amino acids [13]. However, it is highly necessary at this point to think about the acceptability of these wild legumes by farmers and consumers before any further research is conducted in order to orient the improvement, adoption, and domestication for a proper exploitation to the benefit of mankind.

This study explores experienced legumes, cultivating farmers' awareness, perception, acceptability, and preferred uses for some accession of wild *Vigna* legumes (*Vigna racemosa*, *Vigna ambacensis*, *Vigna reticulata*, and *Vigna vexillata*). The study has been organized into two parts, considering farmers' awareness in the first part and preferences for wild legumes in the second.

## 2. Materials and Methods

### 2.1. Study I: Explorative Survey

The aim of this study was to ascertain farmers' awareness about the existence of wild non-domesticated legumes and their uses in addition to challenges and experiences related to the growth and domestication of wild legumes.

The study was conducted among legume farmers in a mid-altitude agro-ecological zone (Arusha Region) and a high altitude agro-ecological zone (Kilimanjaro Region) of Tanzania where legume cultivation is intensified, as shown in Figure 1A [14]. A purposive sampling from a crop-growing population of 0.13% (37,985) from Arusha [15] and 0.17% (56,710) from Kilimanjaro [16] were used to obtain a representative sample size. The total number of farmers involved in legume improvement programs included 50 from the Seliani Agricultural Institute (TARI), Arusha and 100 from the Tanzania Coffee Research Institute (TaCRI), Moshi, Kilimanjaro regions, respectively (Figure 1B). A systematic selection of farmers who had at least two years of trying locally improved legume varieties was performed. An individual face-to-face interview with the help of a semi-structured questionnaire prior to participant experimental plot visit was executed to obtain a broad range of individual opinions and explore their awareness of wild legumes. The questionnaire consisted of 24 items including sociodemographic characteristics. The items were categorized and analyzed to assess the sociodemographic characteristics of participants, their prior knowledge/awareness about wild legumes, and the uses of wild legumes as known by experienced farmers as well as some challenges faced by legume farmers.

### 2.2. Study II: Farmers' Preferences and Perceptions of Wild Vigna Legumes

The main aim of this study was to identify farmers' perceptions and prospective uses of preferred accessions of wild legumes based on morphological agronomic characteristics in order to direct the domestication process.

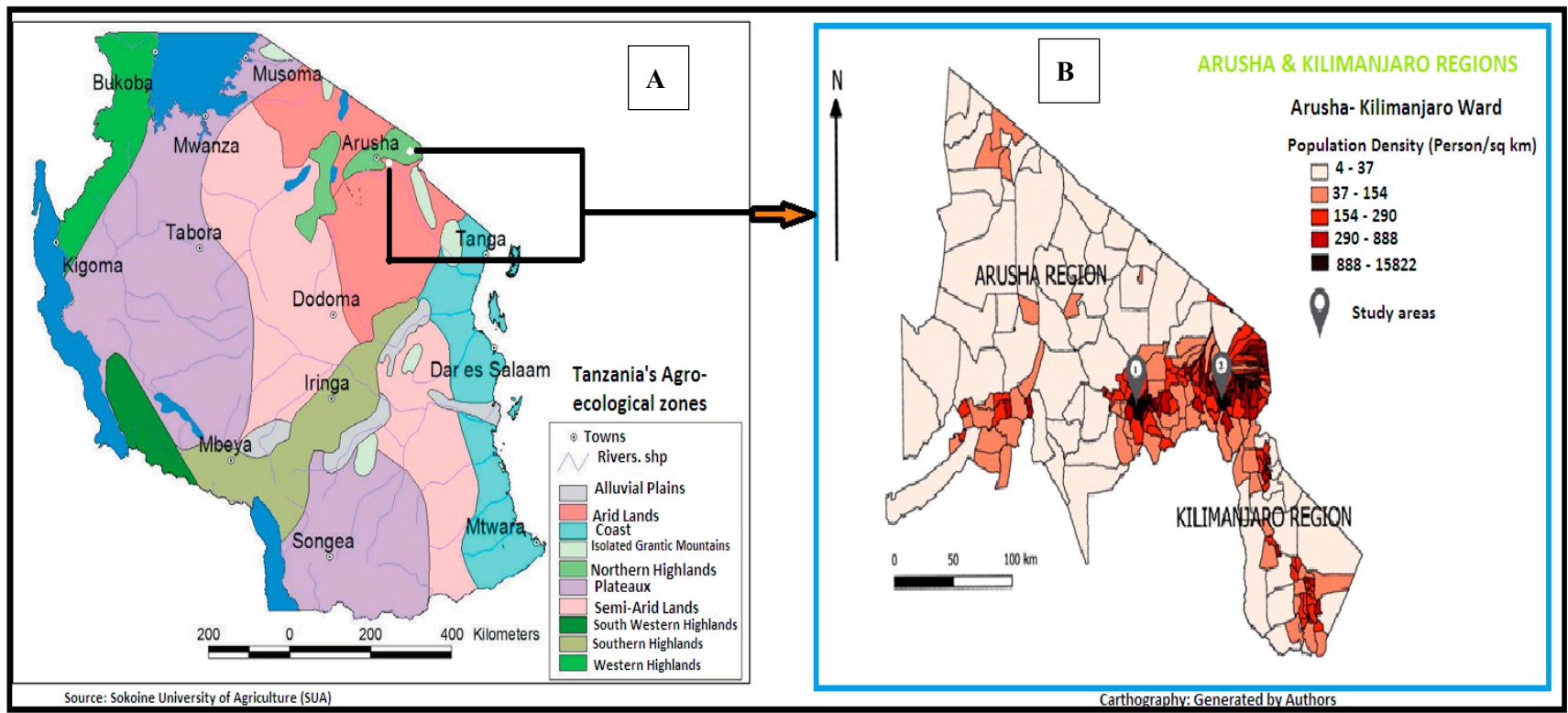

**Figure 1.** Tanzania map showing agro-ecological zones of Tanzania (**A**) [17] and the study sites (**B**): 1 = Arusha district (Arusha region) and 2 = Hai district (Kilimanjaro region).

**Table 1.** Wild *Vigna* species collected from the gene banks/self.

| *Vigna* Species | Genebank/ Number of Accession | | | Total |
|---|---|---|---|---|
| | GRC, IITAIbadan, Nigeria | AGGHorsham, Victoria | Self-Collected | |
| *Vigna racemosa* | - | 4 | - | 4 |
| *Vigna reticulata* | 48 | 3 | - | 51 |
| *Vigna vexillata* | 47 | 13 | - | 60 |
| *Vigna ambacensis* | 42 | 0 | - | 42 |
| *Unknown V. racemosa accession (Nigeria)* | - | - | 1 | 1 |
| *Unknown V. reticulata Accession (Nigeria)* | - | - | 1 | 1 |
| *Unknown Vigna (Tanzania)* | - | - | 1 | 1 |
| Total | 137 | 20 | 3 | 160 |

GRC, IITA: Genetic Resource Center, Germplasm Health Unit, International Institute of Tropical Agriculture (IITA), Headquaters, PMB 5320, Oyo Road, Idi-Oshe, Ibadan-Nigeria. AGG: Australian Grain Genebank, Department of Economic Development, Jobs, Transport and Resources, Private Bag 260, Horsham, Victoria 3401.

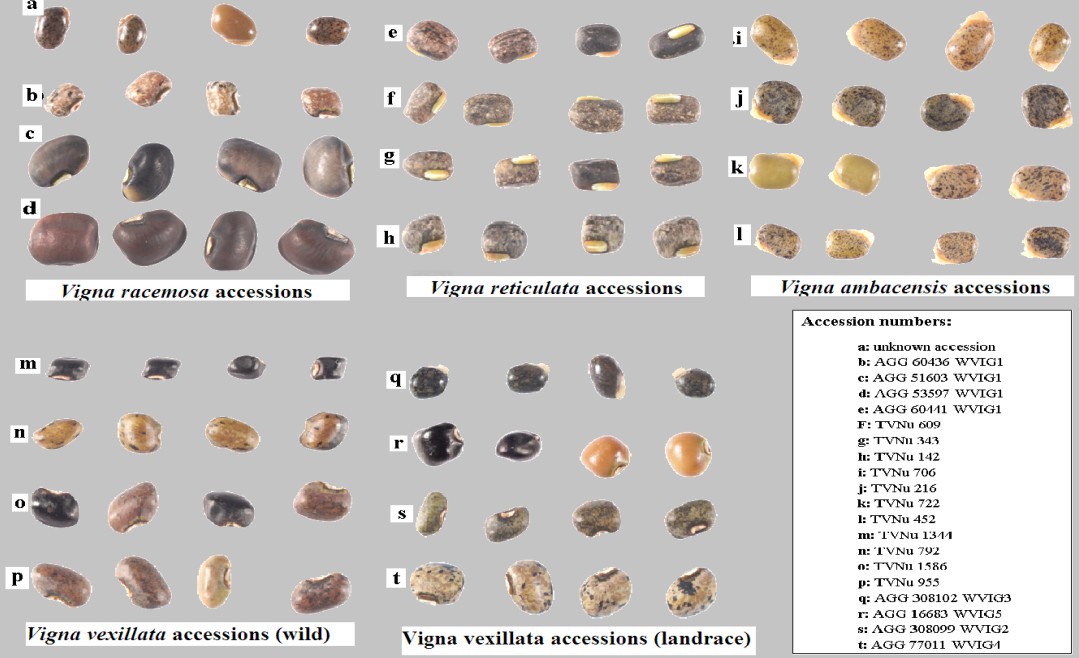

**Figure 2.** Microphotographs illustrating seed morphology of some wild *Vigna* species; Four (4) seeds per accession were pictured under the same conditions to give an image of the morphology and the relative size. Distances of lines in the background are 1 cm in the vertical and horizontal directions. Source: Images taken and compiled by the authors based on seeds requested from the Australian Grain Genebank (AGG) (**a**–**e**,**q**–**t**) and the Genetic Resources Center, International Institute of Tropical Agriculture, (IITA), Ibadan-Nigeria (**f**–**p**).

### 2.2.1. Sample Collection

One hundred and sixty (160) accessions of wild *Vigna* species of legume were obtained from gene banks as presented in Table 1 with their details in Appendix A. All the accessions were planted in an experimental plot following the augmented block design arrangement [18] and allowed to grow until near maturity before inviting the farmers to explore their opinions. Since the accessions did not show uniform growth patterns due to their genetic differences, farmers were invited when more than 50% of the accessions reached maturity. An illustration of the seeds of some of the samples is also shown in Figure 2. In addition, three domesticated *Vigna* legumes—that is, cowpea (*V. unguiculata*), rice bean

(*V. umbellata*), and a semi-domesticated landrace (*V. vexillata*)—were used as checks. The checks were also obtained from the Genetic Resource Center (GRC-IITA), Nigeria, the National Bureau of Plant Genetic Resources (NBPGR), India and the Australian Grain Genebank (AGG), Australia respectively.

### 2.2.2. Experimental Design and Study Site

The study was conducted in two agro-ecological zones located at two research stations in Tanzania during the main cropping season (March-September 2018). One was at the TaCRI, located at Hai district, Moshi, Kilimanjaro region (latitude 3°13′59.59″ S, longitude 37°14′54″ E). The site is at an elevation of 1681 m above sea level, with a mean annual rainfall of 1200 mm and mean maximum and minimum temperatures of 21.7 °C and 13.6 °C, respectively. The second site was at the Tanzania Agricultural Research Institute (TARI), Selian Arusha in the northern part of Tanzania. TARI-Selian lies at latitude 3°21′50.08″ N and longitude 36°38′06.29″ E at an elevation of 1390 m above sea level (a.s.l.) with mean annual rainfall of 870 mm. The mean maximum and minimum temperatures ranged from 22 °C to 28 °C and 12 °C to 15 °C, respectively.

The 160 accessions of wild *Vigna* legumes were planted in an augmented block design field layout following the randomization generated by the statistical tool on the website (http://www.iasri.res.in/design/Augmented%20Designs/home.htm) [19] for 160 treatments with three checks. The field was monitored and maintained in good conditions from germination to near maturity of 75% of all the accessions before inviting farmers to assess their opinions.

### 2.2.3. Participants and Data Collection

Participants in the previous study (Study 1) in the Arusha ($N_1 = 50$) and Kilimanjaro ($N_2 = 100$) regions also participated in this study. Field visits were done in groups of five participants. A trained research assistant was recruited to guide the participants around the experimental field from the first to the last block or vice versa. A semi-structured questionnaire was used to collect information on the most preferred accessions (at least 10), and reasons for each selection were given. Every accession was assigned a number to ease participant selection. The number of times each accession was selected was divided by the total number of selections and multiplied by 100 to give the percentage of selection of each accession.

### 2.2.4. Focus Group Discussion

Participants in their respective regions were further grouped into two groups based on their gender, men and women, giving a total of four group interviews. Each group was invited to participate in an animated video-recorded focus group interview to ascertain their opinion about wild *Vigna* legumes, as obtained in the previous studies. The recorded videos (04) were transcribed verbatim and translated from Swahili language to English. The transcripts were cross-checked with the recordings by the interviewers to align transcripts with notes on non-verbal responses. A coding framework was developed based on the interview objectives and the interview guide. The qualitative data analysis package NVivo 11 (QRS International, 2015) was used to code and organize the data systematically as described by other workers [12]. Key concepts and categories were identified.

### 2.3. Data Analysis

For study I, the collected information during the survey was grouped, coded, organized, and analyzed using the statistical package IBM SPSS Statistic 20.0 (New York, NY, USA). Analysis consisted of the descriptive statistics as well as the binary logistic regression to test for the relationship between the prior knowledge about the wild *Vigna* legumes and the farmers' sociodemographic characteristics.

In the case of study II, data were coded and entered in the statistical package IBM SPSS Statistic 20.0 and analyzed. Analysis included descriptive statistics and likelihood ratio test of $X^2$ to determine the relationship between the preferences and the farmers' gender, farming experience, and research location [20].

## 3. Results

*3.1. Study I*

### 3.1.1. Sociodemographic Characteristics of Participants

The results from the sociodemographic characteristics showed that 64% and 36% were female and male farmers, respectively (Figure 3a). Most of the participants were above 45 years old, with the highest level of education being primary (Kilimanjaro) and secondary (Arusha). Furthermore, most of the farmers had a reasonable number of years of experience farming legumes, varying from two to more than 35 years of farming (Figure 3d). The intervals of years of farming experience and the percentages of participants with the longest farming experience were 6–10 and 16–20%, respectively (Figure 3d).

### 3.1.2. Prior Knowledge/Awareness about Wild Legumes

Less than 30% (28% and 26% in both study sites) of the experienced participants involved in the study were aware of the existence of wild legumes (Figure 4). According to the binary logistic regression analysis (Table 2), the model including the farmers' sociodemographic characteristics as explanatory variables and prior knowledge of legumes as a dependent variable is a good fit with the data as $p = 0.633 > 0.05$ (*Hosmer and Lemeshow test*). This explains that the variance in the outcome is significant ($X^2 = 40.632$, df = 19, p.003) (Omnibus Tests of Model Coefficients). The results show that there is no significant association between the prior knowledge about wild legumes and the overall gender (Wald = 0.495, df = 1, $p > 0.05$) (Table 2). However, there is a slightly effect associated with being a female farmer and prior knowledge (B = 0.303, $p = 0.482$). No significant relationship existed between the overall farmers' age groups and their prior knowledge of wild legumes (Wald = 7.061, df = 6, $p = 0.315 > 0.05$), although there is a slight significance relationship with the youngest age group [15–20] (Wald = 4.113, df = 1, B = 2.982, $p = 0.043$), as shown in Table 2. In the same vein, the test shows that the education level (Wald = 3.962, df = 4, $p = 0.411$) as well as their farming experience (Wald = 5.462, df = 7, $p = 0.604$) do not have any influence to their prior knowledge about wild legumes. On the contrary, the location (research site) has a significant effect on their prior knowledge of wild legumes (Wald = 9.884, df = 1, B = 1.687, $p = 0.002$).

### 3.1.3. Prior Uses of Wild Legumes

A few participants who had prior knowledge of wild legumes mentioned several uses attributed to the wild legumes they had seen before. Some of the uses mentioned were livestock feed, human food, and soil fertility ingredients as well as botanical pesticides (Table 3).

### 3.1.4. Challenges Faced by Legume Farmers

Diseases and drought (or reduced rainfall) were the most challenges faced by the farmers in both mid and high altitude agro-ecological zones (Figure 5). Apart from diseases and reduced rainfall issues, other reported challenges were related to market, pest, and storage (Figure 5). Taste and cooking aspects were not of very serious concern to the farmers in the two zones, since most of them seemed to be comfortable with the taste and cooking aspects of their legumes.

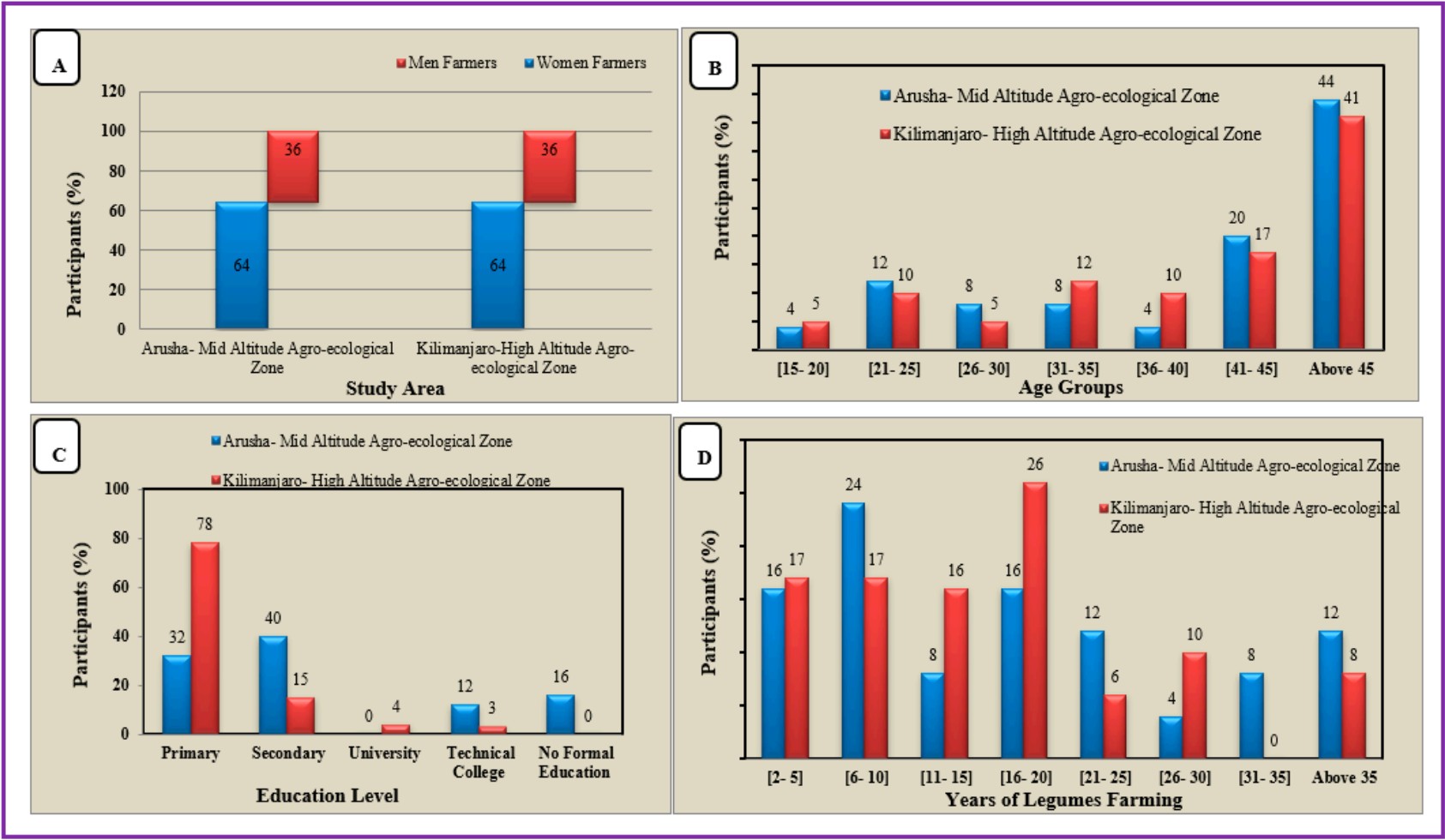

**Figure 3.** Sociodemographic characteristics of participants (**A**): participants' gender per study area (%); (**B**): participants' age groups; (**C**): participants' education level; and (**D**): participants' legumes farming experience.

### 3.2. Study II

#### 3.2.1. Farmers' Preferred Accessions of Wild *Vigna* Legumes

The study shows that 74 accessions out of the 160 planted and grew to an appreciable level at the screening moment and were selected based on the participants' personal preferences (Figure 6). In the high-altitude zone (Kilimanjaro), only five (5) accessions (TVNu-293, TVNu-758, AGG308107WVIG 2, AGG308101WVIG 1, and TVNu-1546) were selected by the farmers more than half of the time, while in the mid-altitude zone (Arusha), none of the accessions had up to 50% selection (Figure 6). The five most selected accessions in the mid-altitude zone—TVNu-293 (36%), TVNu-758 (36%), AGG51603WVIG 1 (30%), AGG308099WVIG 2 (40%), and AGG53597WVIG 1 (34%)—were different from those selected in the high-altitude zone, except for TVNu-293 and TVNu-758.

The likelihood ratio test revealed that the wild *Vigna* selection (preferences) significantly depended on the farmers' gender ($G^2 = 130.813$, df = 73, $p < 0.000$), farming experience ($G^2 = 669.196$, df = 511, $p < 0.000$), and location ($G^2 = 1110.606$, df = 73, $p < 0.000$).

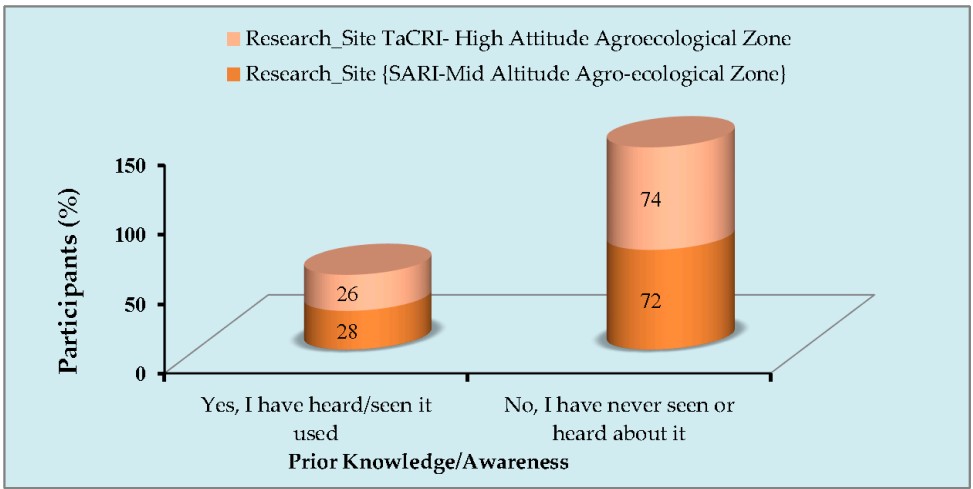

**Figure 4.** Participants' prior knowledge of wild legumes. TaCRI: Tanzania Coffee Research Institute.

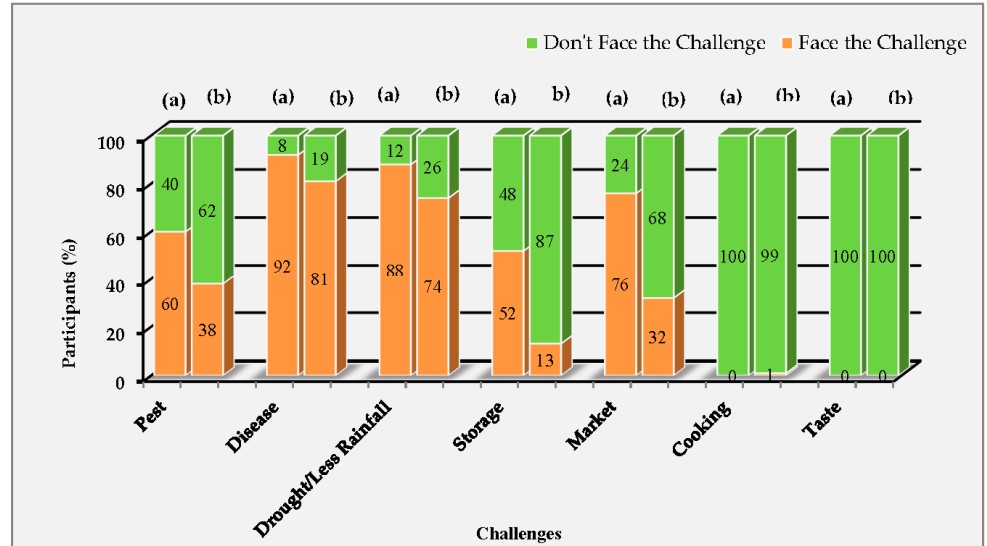

**Figure 5.** Participants' challenges faced during legumes cultivation in the two study areas: (**a**) Arusha and (**b**) Kilimanjaro.

**Table 2.** Binary logistic analysis result.

| | | B | S.E. | Wald | df | Sig. | Exp(B) | 95% C.I. for EXP(B) | |
|---|---|---|---|---|---|---|---|---|---|
| | | | | | | | | Lower | Upper |
| **Variables in the Equation** | | | | | | | | | |
| Step 1 [a] | **Gender (1)** | 0.303 | 0.431 | 0.495 | 1 | 0.482 | 1.354 | 0.582 | 3.153 |
| | **Age** | | | 7.061 | 6 | 0.315 | | | |
| | Age (1) | 2.982 | 1.471 | 4.113 | 1 | 0.043 | 19.732 | 1.105 | 352.281 |
| | Age (2) | 1.162 | 1.010 | 1.325 | 1 | 0.250 | 3.197 | 0.442 | 23.123 |
| | Age (3) | 1.755 | 1.124 | 2.440 | 1 | 0.118 | 5.786 | 0.639 | 52.342 |
| | Age (4) | 1.154 | 0.876 | 1.733 | 1 | 0.188 | 3.171 | 0.569 | 17.668 |
| | Age (5) | 1.010 | 0.798 | 1.601 | 1 | 0.206 | 2.745 | 0.575 | 13.111 |
| | Age (6) | −0.255 | 0.622 | 0.168 | 1 | 0.681 | 0.775 | 0.229 | 2.620 |
| | **Education_Level** | | | 3.962 | 4 | 0.411 | | | |
| | Level (1) | 1.817 | 1.269 | 2.049 | 1 | 0.152 | 6.155 | 0.511 | 74.087 |
| | Level (2) | 2.334 | 1.285 | 3.299 | 1 | 0.069 | 10.316 | 0.831 | 127.995 |
| | Level (3) | 1.694 | 1.763 | 0.923 | 1 | 0.337 | 5.439 | 0.172 | 172.291 |
| | Level (4) | 1.407 | 1.504 | 0.876 | 1 | 0.349 | 4.084 | 0.214 | 77.805 |
| | **Research_Site (1)** | 1.687 | 0.537 | 9.884 | 1 | 0.002 | 5.402 | 1.887 | 15.460 |
| | **Farming_Experience** | | | 5.462 | 7 | 0.604 | | | |
| | Experience (1) | −1.005 | 1.216 | 0.683 | 1 | 0.408 | 0.366 | 0.034 | 3.966 |
| | Experience (2) | −1.245 | 1.118 | 1.242 | 1 | 0.265 | 0.288 | 0.032 | 2.573 |
| | _Experience (3) | −1.222 | 1.022 | 1.430 | 1 | 0.232 | 0.295 | 0.040 | 2.183 |
| | _Experience (4) | 0.121 | 0.873 | 0.019 | 1 | 0.890 | 1.129 | 0.204 | 6.248 |
| | _Experience (5) | 0.409 | 1.025 | 0.159 | 1 | 0.690 | 1.505 | 0.202 | 11.216 |
| | _Experience (6) | −0.559 | 0.998 | 0.313 | 1 | 0.576 | 0.572 | 0.081 | 4.046 |
| | _Experience (7) | 21.259 | 194,50.255 | 0.000 | 1 | 0.999 | 1,708,644,034.887 | 0.000 | . |
| | Constant | −2.586 | 1.058 | 5.975 | 1 | 0.015 | 0.075 | | |

[a]. Variable(s) entered on step 1: Gender, Age, Education_Level, Research_Site, Farming_Experience. B: represent the values for the logistic regression equation for predicting the dependent variables from the independent variables; S.E.: Standard errors associated with coefficients; Wald: Wald $X^2$ value; df: Degree of freedom for each of the tests of the coefficients; Sig.: Significance level (p-value); EXP(B): Exponentiation of the coefficients (odd ratios for the predictors); C.I.: Confidence Interval.

**Table 3.** Wild legumes uses as known by participants with prior knowledge of wild legumes. *

| | Percentage (%) | Livestock Feed | Human Food | Soil Fertility Ingredient | Traditional Botanical Pesticides |
|---|---|---|---|---|---|
| **Participants in a mid-altitude agro-ecological zone** | 28 | 12 Animal feed = '*Chakula cha mifugo*', '*chakula cha ng'ombe*' | 16 Human food = '*Chakula cha binadamu*', *Vegetable* = '*Mboga*' | 0 | 0 |
| **Participants in a high-altitude agro-ecological zone** | 26 | 4 Animal feed = '*Chakula cha mifugo*', '*chakula cha ng'ombe*' | 4 Human food = '*chakula cha binadamu*' *Vegetable* = '*Mboga*'. | 14 Rattlepod (*Crotalaria ochroleuca*) = '*Marejea*' used as fertilizer = '*mbolea*', Nourish the soil = '*Hurutubisha ardhi*', Cover crop = '*Kutandaza shambani*' | 4 Pesticide = '*kunyunyuzia shambani*', '*kutengeza dawa ya kunyunyuzia shambani*' |

* words in single quotation marks (' ') are exact expressions given by participants in Swahili, which has been translated.

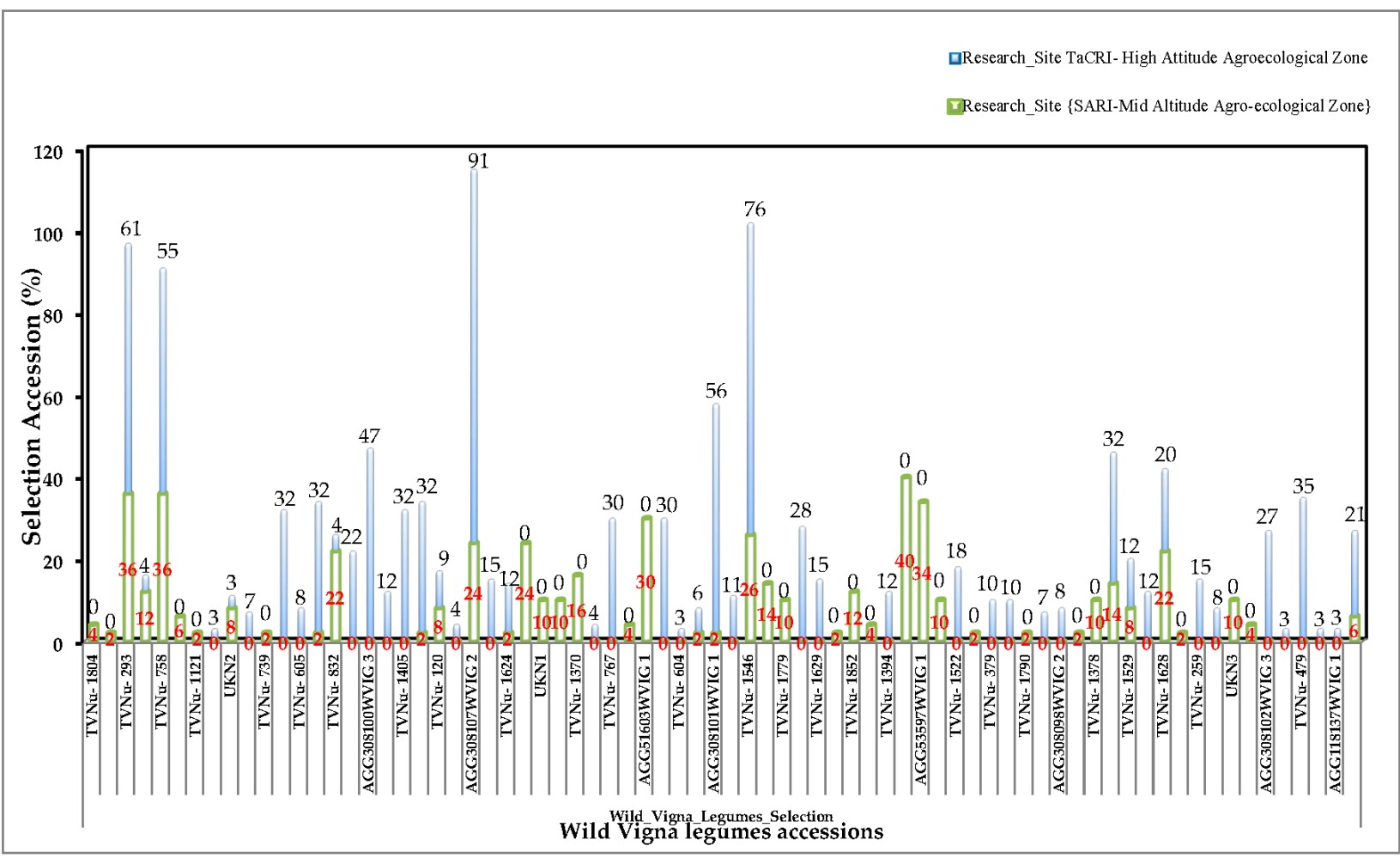

**Figure 6.** Wild *Vigna* legumes preferred (selected) by participants from the two agro-ecological zones.

### 3.2.2. Prospective Uses of Farmers' Preferred Accessions of Wild *Vigna* Legumes

The suggested uses of selected accessions were based on their personal assessment and preferences. Some accessions were selected for more than one use, and the number of selections for every accession is shown on Figure 7a–e. Other uses were proposed by farmers that better suited the accession of their choice. Four main uses (human food, animal feed, forage, and cover crop) were proposed as a result of farmer's preferences and perceptions. Therefore, a total of 31 accessions were preferred as human food (Figure 7a), 49 were preferred as animal feed (Figure 7b), 27 were preferred as forage (Figure 7c), 28 were preferred as cover crop (Figure 7d), and 44 were given specific personal uses (Figure 7e), respectively.

Four accessions were selected at least 30 times or more as human food, while 27 accessions were selected less than 30 times for the same purpose (Figure 7a). The four most selected accessions for this purpose were TVNu-1359 (36), AGG308099WVIG 2 (34), AGG53597WVIG 1 (32), and AGG51603WVIG 1 (30), respectively.

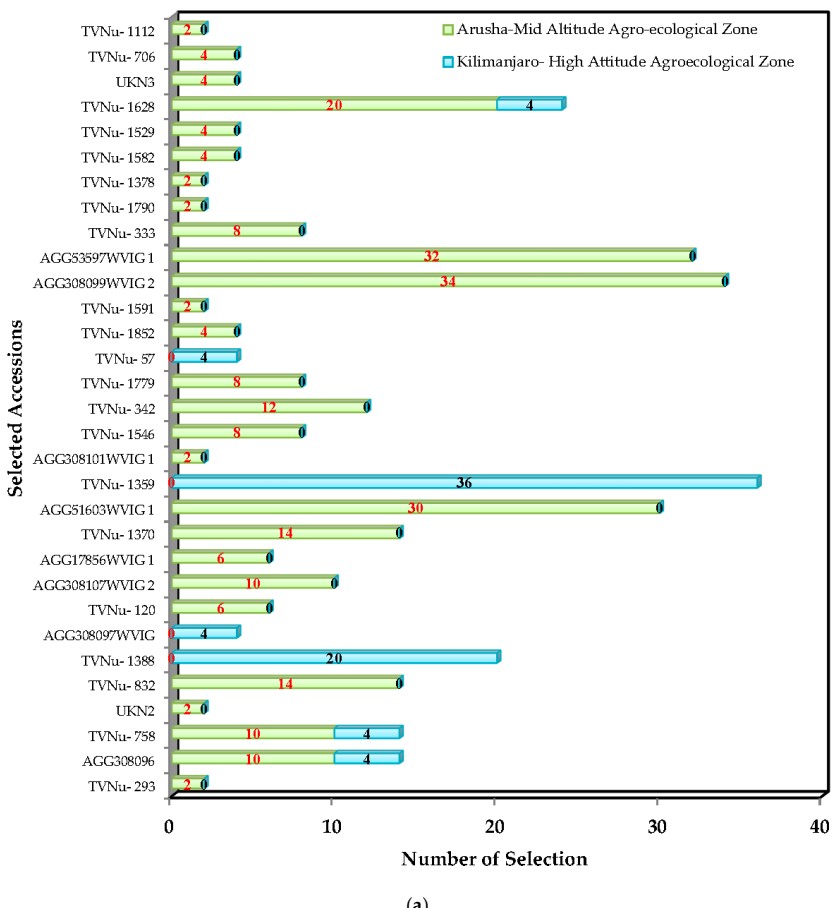

(a)

**Figure 7.** *Cont.*

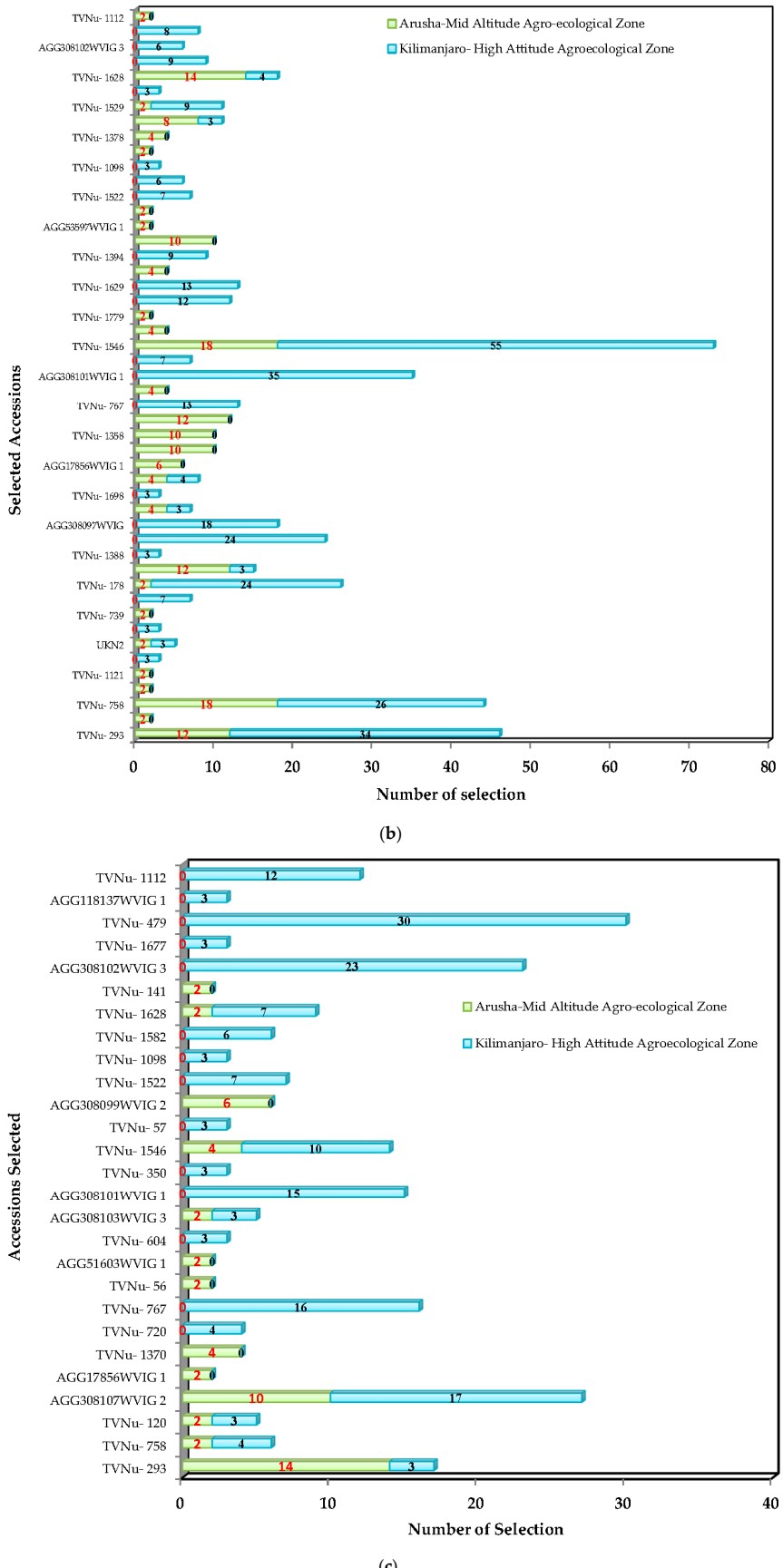

(**b**)

(**c**)

**Figure 7.** *Cont.*

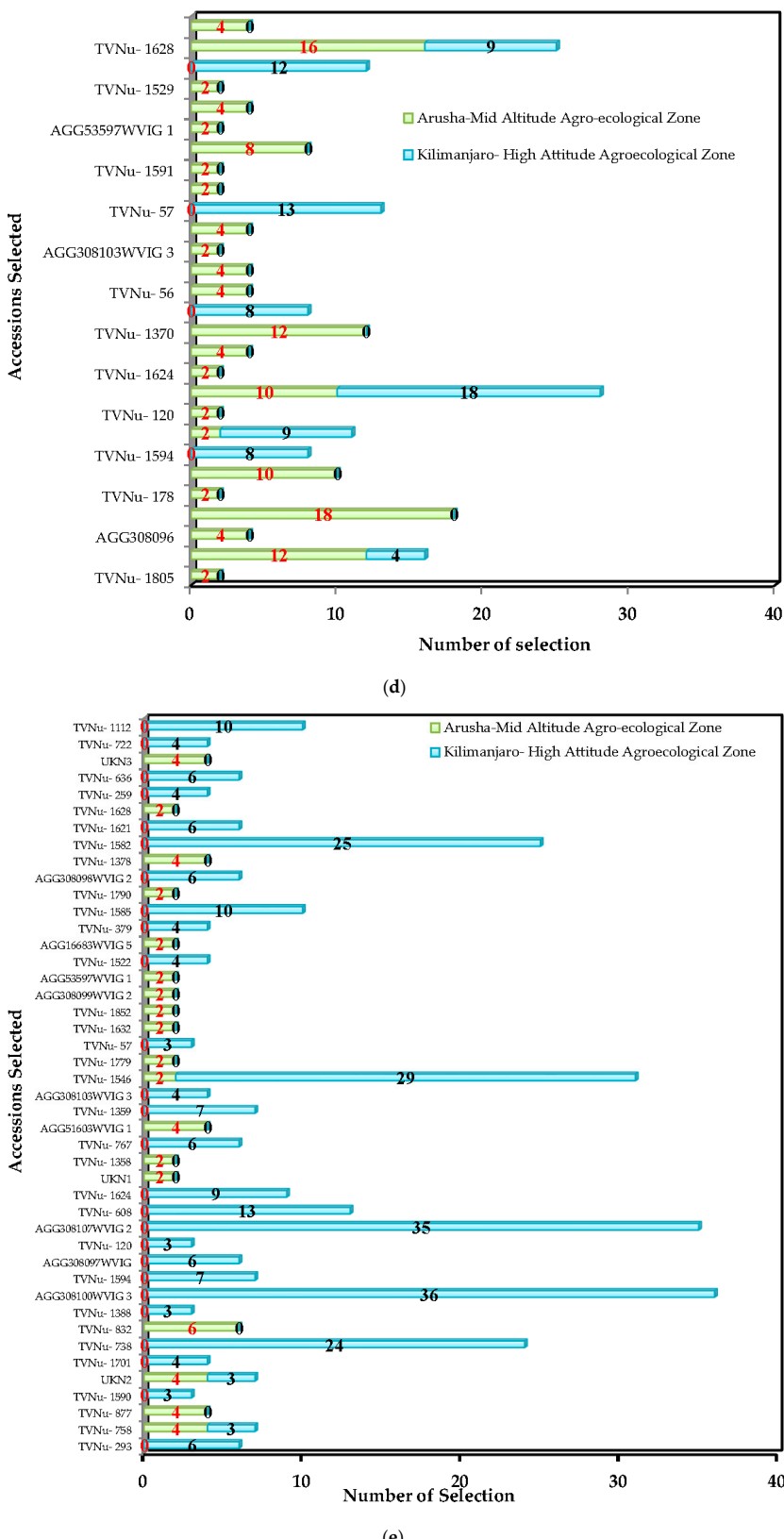

**Figure 7.** (**a**) Wild *Vigna* legumes suggested as a human food; (**b**) Wild *Vigna* legumes suggested as animal feed; (**c**) Wild *Vigna* legumes suggested as forage; (**d**) Wild *Vigna* legumes suggested as cover crop; and (**e**) Wild *Vigna* legumes given specified uses.

Four other accessions were also selected at least 30 times or more by participants as animal feed in the two study sites combined. The selected accessions were TVNu-1546 (18 + 55), TVNu-293 (12 + 34), TVNu-758 (18 + 26), and AGG308101WVIG 1 (35), respectively (Figure 7b).

Only one accession was selected up to 30 times to serve as forage (Figure 7c), while none of the preferred as cover crop accessions were chosen up to 30 times by the participants in both study sites (Figure 7d).

Out of the 44 selected accessions with specified uses, only two accessions—AGG308107WVIG 2 (35) and AGG308100WVIG 3 (36)—were selected more than 30 times (Figure 7e).

All of the non-domesticated wild *Vigna* legumes subjected to this study belonged to four species, *V. racemosa*, *V. reticulata*, *V. vexillata*, and *V. ambacensis*. In summary, it has been shown that the *V. vexillata* accessions were more preferred, followed by *V. reticulata* and *V. racemosa* (Figure 8). Despite the higher number of *V. ambacensis* accessions as compared with *V. racemosa*, it was less selected than *V. racemosa*.

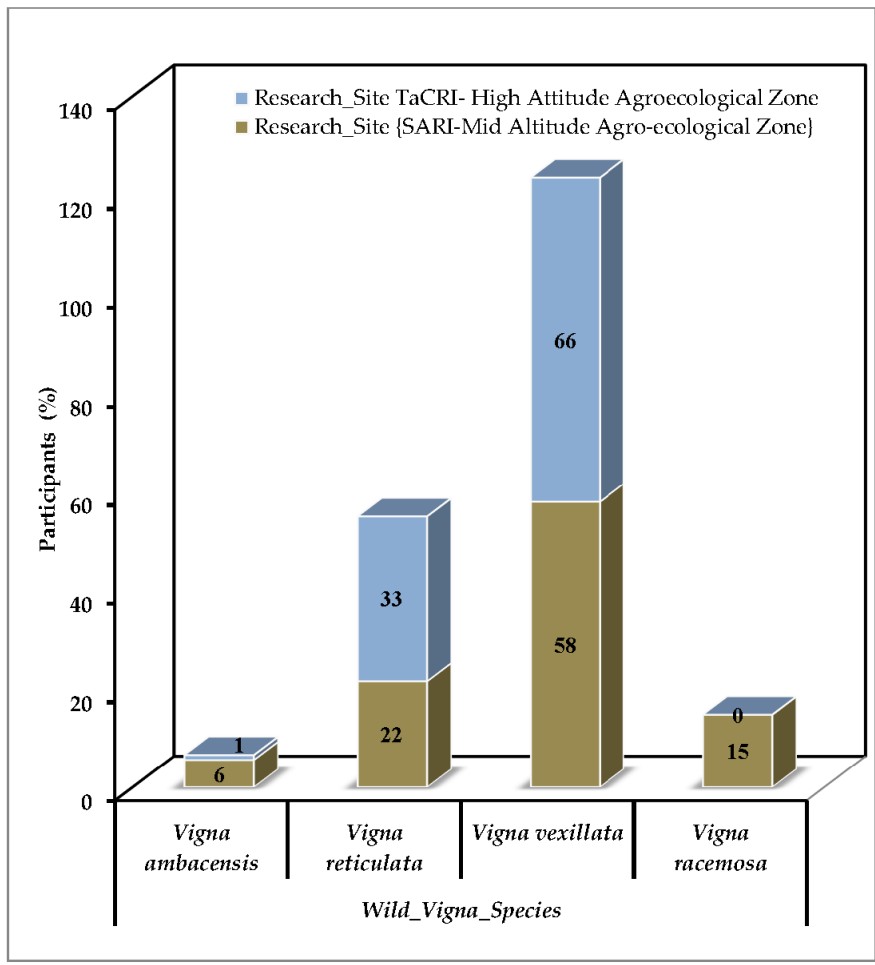

**Figure 8.** Wild *Vigna* legumes selected according to their species.

From their sight and appraisal of the wild *Vigna* legumes, other uses could be organic manure (locally known as '*Mbolea*'—fertilizer), business use, medicinal uses, preventers of soil erosion, and vegetable food for accessions with nice leaves (Figure 9). For personal uses, none of the accessions was selected up to 30 times or more. However, five accessions were selected more than 20 times at least for a specific use. The selected accessions were AGG308100WVIG 3 (24) and TVNu-738 (24) for soil erosion mitigation, and TVNu-1582 (22), TVNu-1546 (26), and AGG308107WVIG 2 (28) for soil fertility as an organic manure agent, respectively (Figure 9).

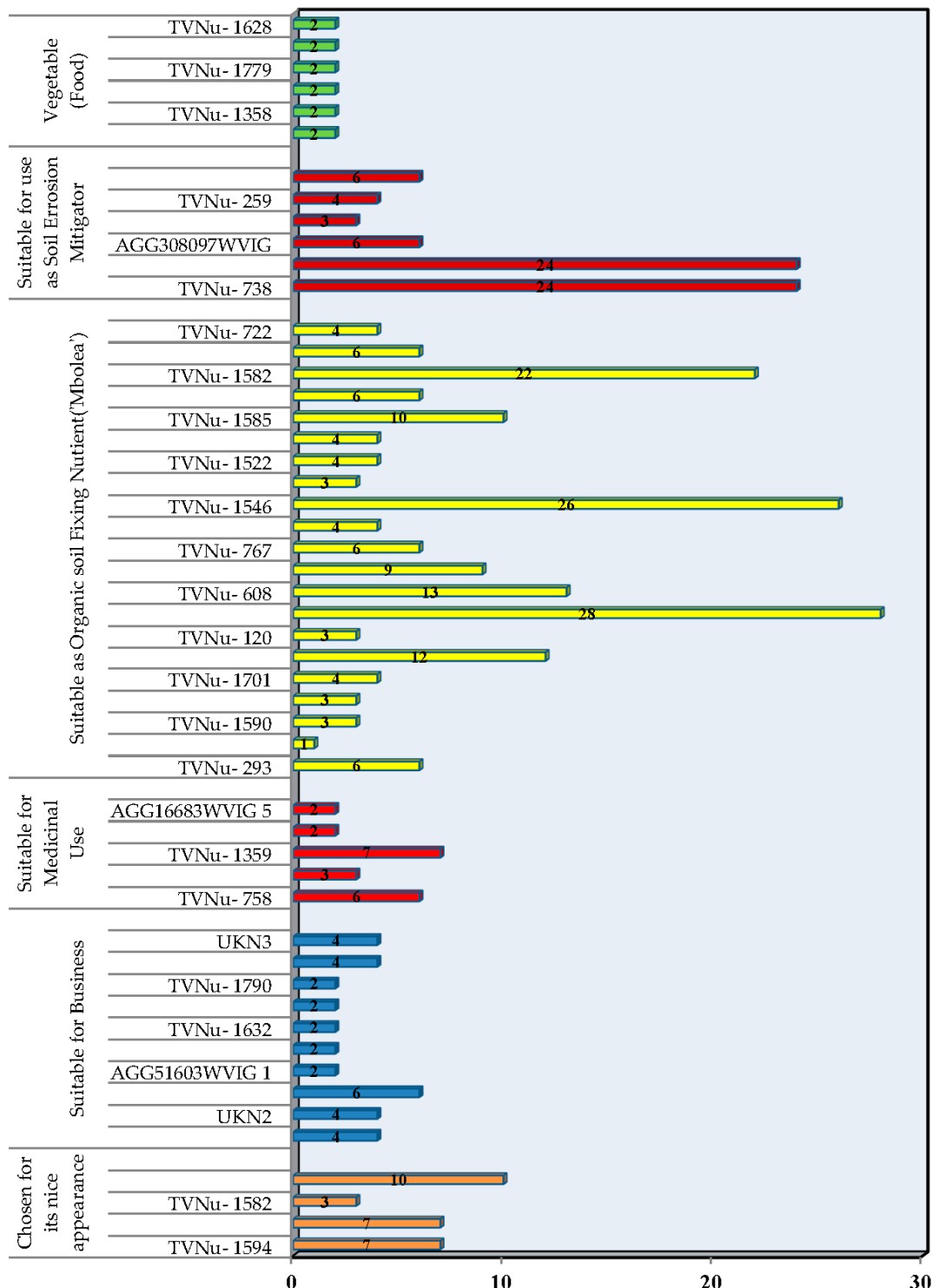

**Figure 9.** Specified uses of wild *Vigna* legumes as proposed by farmers in the two agro-ecological zones of Tanzania.

### 3.2.3. Farmers' Perception of Wild *Vigna* Legumes

From the focus group discussion, most farmers perceived some accessions of wild *Vigna* legumes as good material for future promising business in the field of agriculture due to their high seed production, resistance to drought conditions, and high production of leaves, which can benefit both humans and animals as forage. For example, a male farmer from the Arusha region during the group

discussion enthusiastically responded when the interviewer asked whether they would be willing to adopt some of the wild *Vigna* legumes presented to them for the first time. He said: "Yes, I like some of these beans because many people don't know about them, they are found in the bush but people don't know that they can be useful, so if we discover their usefulness, this can be a great source of good business because they seem to have a higher productivity as compared with other known beans." To support the view, another voice rose in the hall and said: "Yes, I also like some, because after seeing these crops planted in the farm (referring to the wild legumes of study), I discovered that there are other new varieties of legumes, and this may be another source of food. I also realized that some of them have nice leaves that can be used as vegetables, and some can help us feed our cattle."

A smaller proportion of farmers (represented by 26% and 28% in study I, as shown in Figure 3), who curiously noticed the existence of wild legumes before the study, confirmed having seen some of the planted legumes of the study and having consumed them or used them as medicine for animals and even humans. One of the most interesting views that supported this point was from one of the old female farmers in the Arusha region, who said: "This variety with [a] large number of leaves lying on the ground (referring to one of the varieties of the study with a spreading growth habit), I have consumed them several times when I was a kid. Back then, our mothers used to go to the bush and harvest their leaves, and then go to town and buy maize and come back to cook them together. Myself, I have eaten them and we used to call that meal {*Ngolowo*}, which is very delicious and when we mix it with milk, it looks similar to another meal called {*Rojo*}. So for that one, it is not a poison, because I have eaten it before, it is a food, the leaves are eaten and the seeds are also eaten; it is called {*Ngolowo*}". All her mates in the hall during the group discussion listened to her speech with very attuned ears and clapped at the end. A similar view came from the group of men, which was articulated in these terms: "I have seen these beans before growing in the bush and we were using them as food and feed for animals; then, when I saw it here, I just confirmed that it is edible. Animals enjoy them so much. We used to take them from the bush and consume them and we had no health problems with them, and after I saw it here in the farm, I just realized that it is a normal food. It has never affected our people negatively after consuming them.

However, most of the participants in general proposed that more research and improvements were needed, especially in terms of the toxicity and nutritional benefits, as well as the seed color of the legumes to increase their acceptability for efficient exploitation and utilization. "One of the varieties I saw in the farm numbered 132 looks nice; it looks similar to (Choroko, Swahili word for Mung bean). So, I think that if it can be improved, it will be good for business because it has high productivity and nice leaves, but we don't know if it is not toxic or can negatively affect our health", said a participant who was supported by another one, who said: "similar to this one (participant showing some seeds harvested from the experimental fields), if the color can be improved, it will be very nice, because people in the market don't like buying black-colored beans. Their reason is that the black-colored seeds turn the cooking water black and that is not preferable for them. The black-colored seed beans are only preferred during hunger seasons; that is, seasons where less rainfall has affected the crop yield in the community."

## 4. Discussion

The explorative survey above shows that women were more engaged in legume farming in the two zones compared with men. Similarly, the contribution of women in agricultural activities is well-known in Africa [21]. In this study, no statistical significance was found between gender influence and prior knowledge about legumes. This means that being a woman or a man does not influence the probability of being aware of wild legumes.

Legume farming was mainly practiced by the older participants (Figure 3b). This indicates that the younger generations in the areas were not very interested in legume-farming activities or farming other crops. In general, belonging to any age group did not influence the prior knowledge about the legumes, due to the long period of disappearance of the wild genotypes, which led to the ignorance of

many generations of people [2,8,11]. However, belonging to the 15 to 20-year-old age group showed a slight influence on the prior knowledge of wild legumes. This may suggest that farmers in this age range may possess some understanding of wild legumes.

The education level of farmers and their farming experience showed no significant influence on their prior knowledge of wild legumes, which meant that being educated or well experienced in farming legumes did not influence the knowledge of wild legumes. This showed that both experienced and non-experienced farmers as well as educated and non-educated farmers might have the same perception and prior background about wild legumes. In addition, it implied that both farming experience and level of education may not be necessary when making policy decisions about the implementation or adoption of a wild legume as a new crop. However, this is in contradiction with other studies carried out using other domesticated crops such as rice and maize [22,23]. Then, it is necessary for further research to try such experiences with other wild crops in other parts of the world to ascertain this fact.

From the results, the location (research site) has a significant effect on the prior knowledge of wild legumes, meaning that being in the Arusha region increased the chance of knowing wild legumes. Decision making regarding the adoption of wild *Vigna* legumes needs to take the location of farmers into consideration. This is in line with earlier reports [24]. This could be explained by the Arusha region being more populated by a certain ethnic group of people (called the Maasai) who are well-known in Tanzania for their indigenous ethno-medical knowledge of plants [25,26]. They are also found in the high-altitude agro-ecological zone (Kilimanjaro), but they are more concentrated in the mid-altitude agro-ecological zone of Arusha [25].

The ignorance of the wild legumes by the majority of participants in the two study sites may be due to the high and long-term distribution of bred, improved, and landrace varieties of legumes that led to the disappearance, rejection, and negligence of the original wild legumes [2]. However, the numerous challenges (biotic, abiotic, and policy) faced by the improved varieties have recently raised scientific concerns [3]. Therefore, it might be important to go back to the wild and investigate other legumes with good characteristics in relation to their acceptability in order to mitigate the global food insecurity challenge, as pointed out by earlier reports [27].

It is noted from this study that despite the high ignorance noted by the majority, the wild legumes are still used for various purposes, including human consumption by a minority. It has also been noted that ignorance or knowledge/awareness of wild legumes significantly depends on the location of the farmers rather than their gender, age group, or farming experience. This could be explained by some ethnic groups of people with significant traditional and indigenous knowledge of plants being concentrated in some parts of the world [25]. Then, it would be wise to carry out more investigation on such legumes in order to domesticate more varieties possessing resistance to the current legumes challenges. From this study, the main challenges experienced by legume farmers in the two study sites were diseases and low rainfall, which might definitely be due to climate change, as it is global challenge [28]. Therefore, alternatives varieties of legumes with resistance to climate variability and diseases would be of great benefit to such similar communities. The study also attempted to screen some accession of choice by the same farmers based on the general appearance, pods, and seeds of some of the wild legumes in order to select varieties for domestication.

Furthermore, it was observed that the prior knowledge about wild legumes is independent of gender, age, education level, and farming experience, but dependent on the farmers' location. However, it is curiously noted that after carefully sighting the wild *Vigna* legumes performing in the field by participants, it is revealed that there is a significant relationship between the farmers' preferences and their gender, farming experience, and location (likelihood ratio test). This could explain that the knowledge of wild legumes increases farmers' attraction and preferences of wild legumes depending on their gender, farming experience, and location. Less than 50% (74 out of 160) of the planted accessions were preferred by farmers in both research sites (Figure 6), showing that several accessions had common preferences depending on the locations. Although this could be

influenced by the number of accessions that reached an appreciable growth level by the selection period, the selection should depend on other parameters such as farmers' gender ($G^2$ = 130.813, df = 73, $p$ < 0.000), farming experience ($G^2$ = 669.196, df = 511, $p$ < 0.000), and location ($G^2$ = 1110.606, df = 73, $p$ < 0.000), as confirmed by the $X^2$ test. In a similar study, significant correlations between preferences of male and female farmers in an on-farm trial indicated that both groups have similar criteria for the selection of rice varieties in India [29]. Experiments investigating farmers' knowledge about unknown or wild food crops are lacking or almost non-existent in the literature [12]. The wild *Vigna* species are not well-known legumes, which could be the reason taxonomic characterizations have still been under investigation by scientists until recently [30].

The ignorance of wild legumes by the majority explains the few uses suggested by the farmers as compared with the uses suggested after field visits to farms with wild *Vigna* legumes (Figure 4 and Table 3). Several uses have been suggested by farmers after sighting the wild *Vigna* legumes in farms, showing their interest and motivation to adopt some of the wild crops for human benefit. This is in accordance with findings from earlier research studies carried out with domesticated legumes possessing characteristics that are not well-known [31,32]. It was observed that the farmers were willing to adopt some of the crops for several human exploitation purposes, although some need more improvement. It is also noted that some farmers even had experience consuming some of the wild *Vigna* legumes. Therefore, farmers generally perceived the wild *Vigna* legumes as exploitable resources for a variety of purposes that lack awareness and scientific attention. A recent report also demonstrated participant eagerness to adopt wild vegetables (duckweed) as human food upon first-time observations from a picture [12].

This study also shows that there is a high probability that any sample of farmers taken in Tanzania and any other region of the world would ignore the existence of wild legumes. Therefore, considering food insecurity levels in the developing world, the dependence on a few accessions of legumes, and the challenges faced by farmers and consumers regarding domesticated legumes, there is a need to further study these un-exploited legumes and orient their utilization. Very limited reports approaching the assessment of participants, farmers, or consumers' perception, appreciation, or adoption of wild plants as human food exist.

## 5. Conclusions

The existence of non-domesticated wild legumes is highly ignored by many farmers despite the presence of a large existing number in the gene banks and bushes around the world. The ignorance of wild legumes is generally not related to the farmers' gender, age group, or farming experience, while it is significantly related to their location. Besides, preferences in some accessions of wild legumes depend on the gender, farming experience, and location. In addition, the discovery of the wild *Vigna* legumes for the first time motivates the attraction of farmers to prefer them for various purposes. Farmers perceived wild *Vigna* legumes as human food, animal feed, medicinal plants, soil enrichment material, and soil erosion-preventing materials. Therefore, it is necessary for the scientific community to give better attention to these so-called alien species in order to improve their agronomic, nutritional, and physiological characteristics with prior consideration of farmers' and consumers' preferences and perception to orient their domestication, as it is the case here.

**Author Contributions:** P.A.N. conceived and designed the experiments; D.V.H. performed the experiments, collected data, analyzed the data, and wrote the first draft of the manuscript; P.B.V. and A.O.M. supervised the research and internally reviewed the manuscript; and P.A.N. made the final internal review and revised the final draft of the manuscript.

**Funding:** This research was partially funded by the Centre for Research, Agricultural Advancement, Teaching Excellence and Sustainability in Food and Nutrition Security (CREATES-FNS) through the Nelson Mandela African Institution of Science and Technology (NM-AIST) under the reference number P220/CAM.16. The research also received funding support from the International Foundation for Science (IFS) through the grant number I-3-B-6203-1.

**Acknowledgments:** This research was partially funded by the Centre for Research, Agricultural Advancement, Teaching Excellence and Sustainability in Food and Nutrition Security (CREATES-FNS) through the Nelson

**Conflicts of Interest:** The authors declare no conflict of interest.

## Appendix A

**Table A1.** Wild *Vigna* legumes accessions used in the study.

| S/N | Accession Number | Species Name | Genebank |
|-----|------------------|--------------|----------|
| 1 | TVNu-313 | *Vigna ambacensis* | GRC, IITA |
| 2 | TVNu-557 | *Vigna ambacensis* | GRC, IITA |
| 3 | TVNu-1186 | *Vigna ambacensis* | GRC, IITA |
| 4 | TVNu-375 | *Vigna ambacensis* | GRC, IITA |
| 5 | TVNu-1212 | *Vigna ambacensis* | GRC, IITA |
| 6 | TVNu-1792 | *Vigna ambacensis* | GRC, IITA |
| 7 | TVNu-947 | *Vigna ambacensis* | GRC, IITA |
| 8 | TVNu-1679 | *Vigna ambacensis* | GRC, IITA |
| 9 | TVNu-1840 | *Vigna ambacensis* | GRC, IITA |
| 10 | TVNu-219 | *Vigna ambacensis* | GRC, IITA |
| 11 | TVNu-720 | *Vigna ambacensis* | GRC, IITA |
| 12 | TVNu-877 | *Vigna ambacensis* | GRC, IITA |
| 13 | TVNu-706 | *Vigna ambacensis* | GRC, IITA |
| 14 | TVNu-216 | *Vigna ambacensis* | GRC, IITA |
| 15 | TVNu-722 | *Vigna ambacensis* | GRC, IITA |
| 16 | TVNu-1631 | *Vigna ambacensis* | GRC, IITA |
| 17 | TVNu-1677 | *Vigna ambacensis* | GRC, IITA |
| 18 | TVNu-1791 | *Vigna ambacensis* | GRC, IITA |
| 19 | TVNu-765 | *Vigna ambacensis* | GRC, IITA |
| 20 | TVNu-1843 | *Vigna ambacensis* | GRC, IITA |
| 21 | TVNu-629 | *Vigna ambacensis* | GRC, IITA |
| 22 | TVNu-452 | *Vigna ambacensis* | GRC, IITA |
| 23 | TVNu-1185 | *Vigna ambacensis* | GRC, IITA |
| 24 | TVNu-342 | *Vigna ambacensis* | GRC, IITA |
| 25 | TVNu-1125 | *Vigna ambacensis* | GRC, IITA |
| 26 | TVNu-1678 | *Vigna ambacensis* | GRC, IITA |
| 27 | TVNu-223 | *Vigna ambacensis* | GRC, IITA |
| 28 | TVNu-1644 | *Vigna ambacensis* | GRC, IITA |
| 29 | TVNu-1781 | *Vigna ambacensis* | GRC, IITA |
| 30 | TVNu-1851 | *Vigna ambacensis* | GRC, IITA |
| 31 | TVNu-1069 | *Vigna ambacensis* | GRC, IITA |
| 32 | TVNu-456 | *Vigna ambacensis* | GRC, IITA |
| 33 | TVNu-148 | *Vigna ambacensis* | GRC, IITA |
| 34 | TVNu-3 | *Vigna ambacensis* | GRC, IITA |
| 35 | TVNu-1827 | *Vigna ambacensis* | GRC, IITA |
| 36 | TVNu-1691 | *Vigna ambacensis* | GRC, IITA |
| 37 | TVNu-1804 | *Vigna ambacensis* | GRC, IITA |
| 38 | TVNu-1699 | *Vigna ambacensis* | GRC, IITA |

**Table A1.** *Cont.*

| S/N | Accession Number | Species Name | Genebank |
|---|---|---|---|
| 39 | TVNu-1184 | *Vigna ambacensis* | GRC, IITA |
| 40 | TVNu-374 | *Vigna ambacensis* | GRC, IITA |
| 41 | TVNu-1150 | *Vigna ambacensis* | GRC, IITA |
| 42 | TVNu-1213 | *Vigna ambacensis* | GRC, IITA |
| 43 | AGG52867WVIG 1 | *Vigna racemosa* | AGG |
| 44 | AGG51603WVIG 1 | *Vigna racemosa* | AGG |
| 45 | AGG53597WVIG 1 | *Vigna racemosa* | AGG |
| 46 | AGG60436WVIG 1 | *Vigna racemosa* | AGG |
| 47 | *Unknown Vigna racemosa* | *Vigna racemosa* | Self-collected |
| 48 | AGG60441WVIG 1 | *Vigna reticulata* | AGG |
| 49 | AGG17856WVIG 1 | *Vigna reticulata* | AGG |
| 50 | AGG118137WVIG 1 | *Vigna reticulata* | AGG |
| 51 | TVNu-259 | *Vigna reticulata* | GRC, IITA |
| 52 | TVNu-302 | *Vigna reticulata* | GRC, IITA |
| 53 | TVNu-161 | *Vigna reticulata* | GRC, IITA |
| 54 | TVNu-1790 | *Vigna reticulata* | GRC, IITA |
| 55 | TVNu-138 | *Vigna reticulata* | GRC, IITA |
| 56 | TVNu-604 | *Vigna reticulata* | GRC, IITA |
| 57 | TVNu-1112 | *Vigna reticulata* | GRC, IITA |
| 58 | TVNu-312 | *Vigna reticulata* | GRC, IITA |
| 59 | TVNu-224 | *Vigna reticulata* | GRC, IITA |
| 60 | TVNu-1394 | *Vigna reticulata* | GRC, IITA |
| 61 | TVNu-995 | *Vigna reticulata* | GRC, IITA |
| 62 | TVNu-1405 | *Vigna reticulata* | GRC, IITA |
| 63 | TVNu-1522 | *Vigna reticulata* | GRC, IITA |
| 64 | TVNu-379 | *Vigna reticulata* | GRC, IITA |
| 65 | TVNu-609 | *Vigna reticulata* | GRC, IITA |
| 66 | TVNu-1191 | *Vigna reticulata* | GRC, IITA |
| 67 | TVNu-766 | *Vigna reticulata* | GRC, IITA |
| 68 | TVNu-343 | *Vigna reticulata* | GRC, IITA |
| 69 | TVNu-349 | *Vigna reticulata* | GRC, IITA |
| 70 | TVNu-916 | *Vigna reticulata* | GRC, IITA |
| 71 | TVNu-758 | *Vigna reticulata* | GRC, IITA |
| 72 | TVNu-491 | *Vigna reticulata* | GRC, IITA |
| 73 | TVNu-767 | *Vigna reticulata* | GRC, IITA |
| 74 | TVNu-608 | *Vigna reticulata* | GRC, IITA |
| 75 | TVNu-1808 | *Vigna reticulata* | GRC, IITA |
| 76 | TVNu-1825 | *Vigna reticulata* | GRC, IITA |
| 77 | TVNu-1852 | *Vigna reticulata* | GRC, IITA |
| 78 | TVNu-1698 | *Vigna reticulata* | GRC, IITA |
| 79 | TVNu-932 | *Vigna reticulata* | GRC, IITA |
| 80 | TVNu-450 | *Vigna reticulata* | GRC, IITA |
| 81 | TVNu-524 | *Vigna reticulata* | GRC, IITA |
| 82 | TVNu-605 | *Vigna reticulata* | GRC, IITA |
| 83 | TVNu-1156 | *Vigna reticulata* | GRC, IITA |
| 84 | TVNu-607 | *Vigna reticulata* | GRC, IITA |
| 85 | TVNu-1779 | *Vigna reticulata* | GRC, IITA |
| 86 | TVNu-325 | *Vigna reticulata* | GRC, IITA |
| 87 | TVNu-324 | *Vigna reticulata* | GRC, IITA |
| 88 | TVNu-57 | *Vigna reticulata* | GRC, IITA |
| 89 | TVNu-56 | *Vigna reticulata* | GRC, IITA |
| 90 | TVNu-1520 | *Vigna reticulata* | GRC, IITA |

**Table A1.** *Cont.*

| S/N | Accession Number | Species Name | Genebank |
|-----|------------------|--------------|----------|
| 91 | TVNu-602 | *Vigna reticulata* | GRC, IITA |
| 92 | TVNu-1388 | *Vigna reticulata* | GRC, IITA |
| 93 | TVNu-141 | *Vigna reticulata* | GRC, IITA |
| 94 | TVNu-738 | *Vigna reticulata* | GRC, IITA |
| 95 | TVNu-739 | *Vigna reticulata* | GRC, IITA |
| 96 | TVNu-350 | *Vigna reticulata* | GRC, IITA |
| 97 | TVNu-142 | *Vigna reticulata* | GRC, IITA |
| 98 | TVNu-1805 | *Vigna reticulata* | GRC, IITA |
| 99 | *Unknown Vigna reticulata* | *Vigna reticulata* | Self-collected |
| 100 | AGG308102WVIG 3 | *Vigna vexillata* | AGG |
| 101 | AGG308105WVIG 2 | *Vigna vexillata* | AGG |
| 102 | AGG308098WVIG 2 | *Vigna vexillata* | AGG |
| 103 | AGG16683WVIG 5 | *Vigna vexillata* | AGG |
| 104 | AGG308099WVIG 2 | *Vigna vexillata* | AGG |
| 105 | AGG308097WVIG 1 | *Vigna vexillata* | AGG |
| 106 | AGG308101WVIG 1 | *Vigna vexillata* | AGG |
| 107 | AGG308100WVIG 3 | *Vigna vexillata* | AGG |
| 108 | AGG58678WVIG 2 | *Vigna vexillata* | AGG |
| 109 | AGG308103WVIG 3 | *Vigna vexillata* | AGG |
| 110 | AGG308107WVIG 2 | *Vigna vexillata* | AGG |
| 111 | AGG308096 WVIG 2 | *Vigna vexillata* | AGG |
| 112 | AGG62154WVIG 1 | *Vigna vexillata* | AGG |
| 113 | TVNu-1098 | *Vigna vexillata* | GRC, IITA |
| 114 | TVNu-1629 | *Vigna vexillata* | GRC, IITA |
| 115 | TVNu-1718 | *Vigna vexillata* | GRC, IITA |
| 116 | TVNu-1590 | *Vigna vexillata* | GRC, IITA |
| 117 | TVNu-1378 | *Vigna vexillata* | GRC, IITA |
| 118 | TVNu-120 | *Vigna vexillata* | GRC, IITA |
| 119 | TVNu-178 | *Vigna vexillata* | GRC, IITA |
| 120 | TVNu-1796 | *Vigna vexillata* | GRC, IITA |
| 121 | TVNu-1529 | *Vigna vexillata* | GRC, IITA |
| 122 | TVNu-1092 | *Vigna vexillata* | GRC, IITA |
| 123 | TVNu-1546 | *Vigna vexillata* | GRC, IITA |
| 124 | TVNu-1370 | *Vigna vexillata* | GRC, IITA |
| 125 | TVNu-1626 | *Vigna vexillata* | GRC, IITA |
| 126 | TVNu-1358 | *Vigna vexillata* | GRC, IITA |
| 127 | TVNu-1624 | *Vigna vexillata* | GRC, IITA |
| 128 | TVNu-1585 | *Vigna vexillata* | GRC, IITA |
| 129 | TVNu-1617 | *Vigna vexillata* | GRC, IITA |
| 130 | TVNu-1621 | *Vigna vexillata* | GRC, IITA |
| 131 | TVNu-479 | *Vigna vexillata* | GRC, IITA |
| 132 | TVNu-1344 | *Vigna vexillata* | GRC, IITA |
| 133 | TVNu-1628 | *Vigna vexillata* | GRC, IITA |
| 134 | TVNu-381 | *Vigna vexillata* | GRC, IITA |
| 135 | TVNu-792 | *Vigna vexillata* | GRC, IITA |
| 136 | TVNu-1586 | *Vigna vexillata* | GRC, IITA |
| 137 | TVNu-1582 | *Vigna vexillata* | GRC, IITA |
| 138 | TVNu-293 | *Vigna vexillata* | GRC, IITA |
| 139 | TVNu-1359 | *Vigna vexillata* | GRC, IITA |
| 140 | TVNu-955 | *Vigna vexillata* | GRC, IITA |
| 141 | TVNu-1591 | *Vigna vexillata* | GRC, IITA |
| 142 | TVNu-1701 | *Vigna vexillata* | GRC, IITA |
| 143 | TVNu-1443 | *Vigna vexillata* | GRC, IITA |

**Table A1.** *Cont.*

| S/N | Accession Number | Species Name | Genebank |
|-----|------------------|--------------|----------|
| 144 | TVNu-832 | *Vigna vexillata* | GRC, IITA |
| 145 | TVNu-1121 | *Vigna vexillata* | GRC, IITA |
| 146 | TVNu-636 | *Vigna vexillata* | GRC, IITA |
| 147 | TVNu-1476 | *Vigna vexillata* | GRC, IITA |
| 148 | TVNu-1748 | *Vigna vexillata* | GRC, IITA |
| 149 | TVNu-781 | *Vigna vexillata* | GRC, IITA |
| 150 | TVNu-969 | *Vigna vexillata* | GRC, IITA |
| 151 | TVNu-1592 | *Vigna vexillata* | GRC, IITA |
| 152 | TVNu-1632 | *Vigna vexillata* | GRC, IITA |
| 153 | TVNu-333 | *Vigna vexillata* | GRC, IITA |
| 154 | TVNu-1360 | *Vigna vexillata* | GRC, IITA |
| 155 | TVNu-1594 | *Vigna vexillata* | GRC, IITA |
| 156 | TVNu-1369 | *Vigna vexillata* | GRC, IITA |
| 157 | TVNu-593 | *Vigna vexillata* | GRC, IITA |
| 158 | TVNu-1593 | *Vigna vexillata* | GRC, IITA |
| 159 | TVNu-837 | *Vigna vexillata* | GRC, IITA |
| 160 | *Unknown* | *Vigna* | Self-collected, NM-AIST, Tanzania |

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
