# Peer review of "Wild Vigna Legumes: Farmers’ Perceptions, Preferences, and Prospective Uses for Human Exploitation"

_agronomy, doi:10.3390/agronomy9060284_

Reviewer 1 Report

The authors submitted a paper dealing with a research of farmer´s perception and prospective uses of wild Vigna species in Africa, namely in two locations in Tanzania. In fact, the content of the paper is more sociological than biological – it depends on editors of Agronomy journal if such contribution fits to the journal scale. Nevertheless,  I found the paper as useful not only for scientific community per se, but mainly as a challenge for science funds donators and policy makers.

The research was well designed, the experimental/biological part in two locations was correctly realized (160 samples of Vigna species grown) and then reviewed on site by farmers. Of course, in classical field research we usually need at least three year field data – but, for this sociological research it is not – I believe- necessary. The data (questionairs) were correctly evaluated and seriously interpreted.  These data and conclusions may serve as very strong backgound/support  for botanical, agronomical and breeding research of wild Vigna species.

Thus, I recommend this paper after minor revisions for publication in the journal Agronomy.

Questions/comments/ suggestions:

·         Key words: Vigna reticulata

·         Figure 1. Tanzanaia map is deformed – it should be improved

·         Figure 1. The colours on both maps should be explained in the legend.

·         Page 5, Table 1: Provide a location/address of gene banks, i.e. towns in particular countries

·         Plate 1 (why not Figure 1?) – Provide a measure/scale of seeds in mm; are they presented in the similar/comparable sizes?

·         Page 6. From biological and methodological point of view is a key question the date of maturation of 160 samples of Vigna for inspection. It is probable, that vegetation period of these samples is not uniform. Please, provide an explanation in all chapters of ms (MM, Results, Discussion).

·         Discussion – The form of discussion is probably more suitable for sociological than for biological paper; nevertheless it is acceptable for me.

·         References must be thoroughly checked and improved (uncomplete citations; many formal mistakes)!

Author Response

RESPONSES TO REVIEWER #1 COMMENTS ON THE MANUSCRIPT SUBMITTED TO AGRONOMY 

The authors would like to first thank you as well as the reviewers for your precious time and invaluable thoughtful comments towards the improvement of our article. We have carefully addressed all the comments requested. The corresponding changes and refinements made in the revised paper are summarized in our responses below. Generally, all the corrected points and changes made in the revised version of the manuscript can easily be tracked as the track changes option of Microsoft Word was activated during the revision. 

 REVIEWER #1 COMMENTS AND RESPONSES:

 Point 1: Reviewer point out a spelling mistake in the section; Key words: Vigna reticulata 
 Response 1: The spelling has been corrected (see line 44 page 2) 

Point 2: Figure 1. Tanzanaia map is deformed – it should be improved Response 2: The Tanzania map has been improved as recommended (see page 4) Point 3:  Figure 1. The colours on both maps should be explained in the legend. Response 3: The colors on both maps have been explained in a legend as recommended (page 4) 

Point 4: Page 5, Table 1: Provide a location/address of gene banks, i.e. towns in particular countries Response 4: The locations of genebanks have been provided as recommended (Page 5, lines 140- 143). 

Point 5: Plate 1 (why not Figure 1?) – Provide a measure/scale of seeds in mm; are they presented in the similar/comparable sizes? Response 5: The plate 1 has been renamed Figure 2 as recommended and the figure numbers adjusted in the whole manuscript as shown on pages 6, 7, 9, 11, 13, 14, 15, 16, 17, 18, 19, 20, and 21by following the track changes. The pictorial views of seeds presented here are actually micrographs of the seeds taken from few seeds just to serve for illustrative purpose only; Four (4) seeds per accession were pictured under the same conditions to give an image of the morphology and the relative size. Distances of lines in the background are 1 cm in vertical and horizontal directions. The additional information has been added accordingly as recommended (see page 8, lines – 147- 150).

Point 6:  Page 6. From biological and methodological point of view is a key question the date of maturation of 160 samples of Vigna for inspection. It is probable, that vegetation period of these samples is not uniform. Please, provide an explanation in all chapters of ms (MM, Results, Discussion). Response 6: Explanations have provided in the following sections as recommended: - Materials and Methods: Explanations have been provided at page 7, lines 134- 135. - Results: Explanations have been provided at page 15, lines 260- 261. - Discussion:  Explanations have been provided at page 26, lines 457- 459. 

Point 7: Discussion – The form of discussion is probably more suitable for sociological than for biological paper; nevertheless it is acceptable for me. biological paper; nevertheless it is acceptable for me. Response 7: Ok Point 8: References must be thoroughly checked and improved (uncomplete citations; many formal mistakes)! Response 8: References have been checked and improved as recommended (see section references, changes are shown in red as tracked by Microsoft Word track changes option). Major changes have been made on the following references: Reference number 5, 9, and 10. Generally, the references have been managed by Mendeley reference manager to avoid some mistakes.

Reviewer 2 Report

The manuscript is well written and will contribute to the future sustainability of food supply. There are some misspelling that need to be corrected.
L68: V. unguiculata (L.) Walp ---> V. unguiculata (L.) Walp.

L69: V. subterranea (L.) Verdn. ----> V. subterranea (L.) Verdc.

L69: V. aconitifolia (Jacq.) ----> V. aconitifolia (Jacq.) Maréchal

L134: V. umbellate ----> V. umbellata

L151 and 155:  There are two study sites, named mid-altitude agro-ecological zone (Arusha reion) and a high altitude agro-ecological zone (Kilimanjaro regio). However, altitude of Kilimanjaro region is 1268 m while that of Arusha is 1390 m. This is strange and need to be checked.

Plate 1.: Seed photo of (q) suggest misidentification. (q) might not be Vigna vexillata. This is not fault by the authors but please check if possible.

L. 542: Norihiko T. ----> Tomooka N.

Author Response

RESPONSES TO REVIEWER #2 COMMENTS ON THE MANUSCRIPT SUBMITTED TO AGRONOMY 

The authors would like to first thank you as well as the reviewers for your precious time and invaluable thoughtful comments towards the improvement of our article. We have carefully addressed all the comments requested. The corresponding changes and refinements made in the revised paper are summarized in our responses below. Generally, all the corrected points and changes made in the revised version of the manuscript can easily be tracked as the track changes option of Microsoft Word was activated during the revision. 

 REVIEWER #2 COMMENTS AND RESPONSES: 

The manuscript is well written and will contribute to the future sustainability of food supply. There are some misspellings that need to be corrected. 

Point 1: L68: V. unguiculata (L.) Walp ---> V. unguiculata (L.) Walp. Response 1: The full scientific name has been corrected as pointed out; a full stop has been added after ‘’Walp.’’ (page 2, line 68) 

Point 2: L69: V. subterranea (L.) Verdn. ----> V. subterranea (L.) Verdc. Response 2: The spelling has been corrected as pointed out (see page 2, line 69) Point 3 : L69: V. aconitifolia (Jacq.) ----> V. aconitifolia (Jacq.) Maréchal Response 3: The species scientific name has been corrected (see page 2, line 69/70) Point 4: L134: V. umbellate ----> V. umbellata Response 4: The spelling has been corrected accordingly (see page 7, line 137) 

Point 5: L151 and 155:  There are two study sites, named mid-altitude agro-ecological zone (Arusha reion) and a high altitude agro-ecological zone (Kilimanjaro regio). However, altitude of Kilimanjaro region is 1268 m while that of Arusha is 1390 m. This is strange and need to be checked. Response 5: The authors acknowledge the vigilance of the reviewer on this point. This came from a typing mistake and might have caused a serious technical and scientific problem to the understanding of reader. The point has been corrected (see page 8, lines 157- 158). The elevation of Hai District (Kilimanjaro region) is actually1681 m above sea level. 

Point 6: Plate 1.: Seed photo of (q) suggest misidentification. (q) might not be Vigna vexillata. This is not fault by the authors but please check if possible. Response 6: The information on the identification of seed photo (q) has been given by the Australian Grain genebank as Vigna vexillata. Authors have checked the passport data given by the gene bank and confirmed that it is actually Vigna vexillata. 

Point 7: L. 542: Norihiko T. ----> Tomooka N. Response 7: The pointed reference has been corrected accordingly (see Reference number 9, line 549 Page 33)
